# Review on the Research and Applications of TLS in Ground Surface and Constructions Deformation Monitoring

**DOI:** 10.3390/s22239179

**Published:** 2022-11-25

**Authors:** Jinlong Teng, Yufeng Shi, Helong Wang, Jiayi Wu

**Affiliations:** College of Civil Engineering, Nanjing Forestry University, Nanjing 210037, China

**Keywords:** TLS, deformation monitoring, point cloud, ground surface, constructions

## Abstract

With the gradual maturity of the terrestrial laser scanners (TLS) technology, it is widely used in the field of deformation monitoring due to its fast, automated, and non-contact data acquisition capabilities. The TLS technology has changed the traditional deformation monitoring mode which relies on single-point monitoring. This paper analyzes the application of TLS in deformation monitoring, especially in the field of ground surface, dam, tunnel, and tall constructions. We divide the methods for obtaining ground surface deformation into two categories: the method based on point cloud distance and the method based on displacement field. The advantages and disadvantages of the four methods (M2M, C2C, C2M, M3C2) based on point cloud distance are analyzed and summarized. The deformation monitoring methods and precisions based on TLS for dams, tunnels, and tall constructions are summarized, as well as the various focuses of different monitoring objects. Additionally, their limitations and development directions in the corresponding fields are analyzed. The error sources of TLS point cloud data and error correction models are discussed. Finally, the limitations and future research directions of TLS in the field of deformation monitoring are presented in detail.

## 1. Introduction

With the development of human activities, there are more and more constructions. At the same time, disasters caused by construction deformation occur from time to time (such as building collapse, landslide, debris flow, tunnel collapse, etc.), causing great losses to human life and property safety. Therefore, the monitoring and analysis of the ground surface and constructions deformation has become increasingly important. It is necessary to regularly monitor the constructions to reasonably ensure and maintain their safety [1]. Continuous observation and analysis of deformation phenomena of deformed bodies by using special instruments and methods, and prediction of deformation trends can avoid serious consequences caused by construction deformations. The traditional deformation monitoring techniques mainly use level, total station, or GNSS (global navigation satellite system) for single-point observation [2]. However, these observation methods have few monitoring points and cannot provide comprehensive monitoring of objects. Moreover, it is difficult to lay control points in complex terrain areas because of the heavy field work and the long period to obtain results, which greatly affects the monitoring efficiency [3,4].

Three-dimensional (3D) laser scanning is an emerging measurement technology, which is used to quickly and accurately capture points on the object surface with its advantages of fast speed, automation, non-contact, and high precision [5]. Its appearance has changed the traditional single-point deformation observation, and it is a whole-based deformation monitoring method. The acquired data points are generally defined based on x, y, z coordinates associated with attributions, such as the intensity of the laser beam reflected from the observed object. Over the last two decades, geomatics experts, researchers, and practitioners have witnessed a dramatic change in the way surveying is conducted. Point clouds are the most viable kind of data, representing various objects at different scales and levels of complexity [6]. During data acquisition, laser scanning units can be classified based on the position of the laser sensors; these classifications are aerial, mobile, and terrestrial laser scanning corresponding to air (e.g., helicopter, plane, or drone), mobile equipment (e.g., vehicle, train, or boat) and the ground [7]. Each method has its own advantages, but TLS does not need a carrier as it can work when mounted on a tripod. TLS technology has been used more commonly and is popular. Therefore, its applications to the field of deformation monitoring has great significance and research ability [8]. TLS also has great potential in the inspection processes due to its ability to capture objects in high speed with accuracy up to sub-millimeter, and its low cost compared to other traditional inspection methods.

In the past decade, a large number of studies have proved the feasibility of TLS in deformation monitoring, and deformation monitoring has been applied in surface, building, dam, tunnel, and other scenes [9]. For the literature search, the main source of information was considered to Web of Science. Moreover, a supplementary literature search was also conducted in Google Scholar and PubMed. We obtained 527 articles by searching for the keywords terrestrial laser and deformation. Keywords are important to present the fundamental concepts/concerns and subject areas of the published work, and demonstrate a quick overview of the research horizons [10]. Co-occurrence of keywords analysis is presented in Figure 1.

In view of the keyword association in Figure 1, we chose to summarize the deformation methods of ground surface, dams, shield tunnels, and tall constructions. In this paper, we review the deformation monitoring methods of ground surface and constructions based on TLS. Section 2 introduces the applications of TLS technology in ground surface and slope monitoring. Section 3 summarizes the applications of TLS in deformation monitoring of constructions (including dams, shield tunnels, and tall constructions). Section 4 analyzes the influencing factors of TLS deformation monitoring, including registration, filtering, and data quality. Section 5 summarizes the whole paper and proposes the future research directions of TLS in deformation monitoring.

## 2. Research and Applications of TLS in Ground Surface Deformation Monitoring

Under the influence of long-term geological processes and human factors, the original stress state of natural or artificial surface slopes will change, leading to stress redistribution and stress concentration, which will cause the surrounding rock and soil to experience different forms and even deformation, thus endangering the safety of nearby infrastructure and people. Therefore, it is important to monitor the safety and stability of slopes. The traditional monitoring methods include manual measurement, total station, GNSS, and close-range photogrammetry [11]. Although TLS generally cannot measure the same point twice as opposed to other techniques, which resulted in a loss of single-point accuracy, deformation monitoring based on TLS can provide discrete 3D data of the surface, avoiding the locality and unilaterality of stress-strain analysis based on single-monitoring-point data. At the same time, it can improve the efficiency, accuracy, and diversity of data types, which ensures that the surface slope deformation monitoring based on TLS has a wide application prospect.

The surface slope deformation monitoring mainly calculates two types of quantities: (i) distance measurement between point clouds based on the variation difference with time; (ii) displacement field calculation based on the identification of corresponding elements in continuous point clouds.

### 2.1. Deformation Monitoring Method Based on Point Cloud Distance

Methods for calculating point cloud distance have been developed for model-to-model (M2M), cloud-to-cloud (C2C) [12], cloud-to-mesh/model (C2M) [13], multi-scale model-to-model cloud comparison (M3C2) [14], and their optimizations. In point cloud distance measurement, for point-based strategies, point cloud is reduced to a subset of salient points, also known as features or key points, which have unique characteristics considering their local neighborhood. This means that they can be found in point clouds of different measurement periods and matched between point clouds without the need to pre-register the point clouds in a common coordinate system and use the neighborhoods in Euclidean space [15]. Grid-based methods require the creation of models from one or two point clouds, respectively, but are more robust to outliers than C2C, which finds the nearest points based only on Euclidean distance [14,16]. In addition, multi-temporal datasets must be registered prior to change analysis, which is usually performed using signal markers or defined stability regions where the difference between the two datasets is minimized [17]. The M3C2 algorithm is currently a common technique for change analysis based on geomorphic point clouds, providing accurate measurements at multiple scales in a simpler workflow that works in two steps: (1) the surface normal estimation in 3D at a scale consistent with the local surface roughness, and (2) the measurement of the mean surface change along the normal direction with explicit calculation of a local confidence interval [18].

Many researchers use the method of point cloud distance measurement to obtain deformation information. In the study of the seasonal deformation on the Qinghai–Tibet plateau, TIN (triangulated irregular network) has the advantage of representing the digital elevation features by topographic feature points, and continuously calculating differences between multi-temporal TINs to obtain the elevation fluctuation of the Qinghai–Tibet plateau [19]. The information on surface changes of a slow-moving landslide in the Austrian Flysch zone was also obtained based on M2M [20]. This method was also applied to the subsidence basin of the Wangjiata mining area, which was obtained by DEM subtraction in two phases, with an accuracy up to 67 mm [21]. The surface DEM is established by using the point clouds after coordinate transformation, and the dynamic subsidence value and subsidence basin during the observation period are obtained by DEM subtraction. When monitoring the fissure of the ground surface, the differences between point cloud registration are used as the global displacement [22]. Due to the complexity and heterogeneity of the ground fissure disaster evolution, it is difficult to obtain a reasonable global registration result to express the characteristics of deformation and damage by best-fitting methods. Therefore, control points are set in the non-deformation area, and point clouds in different periods are registered by using control points to obtain surface subsidence. The C2C method was applied to landslide monitoring to calculate landslide displacement [23]. Then, an upgraded method on C2C was proposed, which used a combination of point distance and statistical sampling to measure deformation directly from point cloud data without mesh or curve fitting to maximize data integrity and minimize errors [16]. C2M is the most common technique in inspection software. Surface change is calculated by the distance between a point cloud and a reference 3D mesh or theoretical model [13,24,25]. The M3C2 algorithm was used to obtain the rock glacier surface deformation and orientation of the landslide at different intervals [26,27]. It found that higher temporal resolution will play an important role in the future observation of glacier rocks, the average distance of boulder movements during the 3-week period is 0.08 m (±23 mm, standard deviation) [26]. However, M3C2 relies on position averaging in the reference and data point clouds to reduce noise, which may cause undesired artifacts in high roughness surfaces or around occluded points. Therefore, a point cloud denoising and calibration method is proposed to solve this problem [17], which used the point redundancy in space and time, calculates the distance using the multi-scale normal distance method, and uses a set of calibrated point clouds to eliminate the systematic error. The median value is used to filter the distance value of the neighborhood in space and time to reduce random type errors and improve the ability of monitoring and detecting small changes. Additionally, a method for change detection and change quantification using 3D point clouds was presented [28], which extends the state-of-the-art M3C2 method by incorporating knowledge about the uncertainty of individual points. It applied the error propagation theory to the M3C2 algorithm and upgraded it to M3C2-EP to reduce the quantization difference of LoDetection (level of detection) in the M3C2 algorithm and the number of false alarms. However, it was difficult to interpret changes measured using the M3C2 method when: (1) change occurs in directions different to the direction of change computation, or (2) the quantified magnitudes of change are exceeded by the associated uncertainty due to a rough surface morphology. The approach named correspondence-driven plane-based M3C2 approach was presented, to quantify the small-magnitude (<0.1 m) 3D topographic change of rough surfaces by reducing the uncertainty of quantified change [29]. The advantages and disadvantages of point cloud ranging methods are compared in Table 1.

### 2.2. Deformation Monitoring Method Based on Displacement Field

It is a common method to construct a 3D displacement field to represent the deformation direction of the surface area. The displacement field contains displacement vectors with corresponding directions, which represent the displacement change trend of the surface of the survey area. The displacement field operation often uses a sliding window to register the regional point cloud to obtain the displacement vector. In other words, the difference in the method based on displacement field is the different way of point cloud registration.

The cross-correlation-based particle image velocimetry (PIV) method was applied to derive a surface deformation field [30]. Point clouds in different periods are gridded on the XY plane; the first stage point clouds were fixed, and the second stage point clouds were registered with the first stage point clouds by using a correlation window and inquiry window and calculating the correlation matrix. The disadvantages are that significant surface damage between continuous scans will reduce the correlation, and a single PIV parameter is not applicable for complex landslide deformation. When extracting landslide displacement, the ICP (iterative closest point) algorithm [31] deduced a 3D displacement vector with a maximum 3D deviation of about ±7 cm [32]. To simplify computation time, the point clouds are divided into unit elements of selectable size and an ICP matching algorithm is applied to each voxel. The correspondence between points and surfaces is calculated [33], and the transformation matrix derived for each voxel is used to connect each point of the primary point cloud with the corresponding surface of the secondary point cloud, and the generated connections can describe the motion between point clouds. A method called PCSC (point cloud set conflict) was proposed for giant landslide displacement analysis to acquire the surface changes from two temporal point clouds [34]. In this method, the surface displacement results, including magnitude and directions, are calculated by the PCSC technique based on terrain roughness. However, the deviations between PCSC and InSAR (interferometric synthetic aperture radar) results are at a centimeter level (1.18 cm). The confidence of a centimeter could be acceptable in study of an oversized landslide. The coarse registration method of point cloud based on SIFT (scale-invariant feature transform) feature points and the fine registration method based on an improved ICP algorithm of K-D tree were applied to slope deformation monitoring [35]. It constructs a prism with the normal vector of the reference point cloud as the axis, determines the position of the comparison point cloud by calculating the center of gravity in the prism, extracts 3D deformation information along the normal line, and determines the optimal model parameters through model tests. The mobile registration window technique is used to determine the displacement field of underground mining. The accuracy of the proposed method to automatically determine the displacement field, when taking into account the georeference error, was within the range of a few centimeters [36]. The technology divides the reference point cloud and the point cloud to be registered into multiple segments. The corresponding segments are then consecutively registered, and for each registration window, the corresponding displacement vector is determined using the transformation parameters between the base segment and the registered point cloud. In the process of registration, the size of the window is critical. Some people have made breakthroughs in the field of deformation monitoring using deep learning. For the previous algorithm, the displacement vector only represents the distance between surfaces, rather than the actual displacement of points on the surface. Therefore, the derived vector does not represent the real 3D displacement, and the result is usually interpreted and visualized only on the basis of its amplitude. Furthermore, it mainly represent deformations orthogonal to the surface, and does not reflect or significantly underestimate the actual displacements in the case of motions/deformations parallel to the surface or more complex motions [37,38]. A novel deep learning framework was proposed to compute realistic 3D displacement vectors, called F2S3 (feature-to-feature supervoxel-based spatial smoothing) [38,39]. The main idea of F2S3 is to establish corresponding points in each period based on the proximity in the feature space spanned by local feature descriptors, rather than the proximity in Euclidean space. The local feature descriptors describe the geometric information of the local neighborhood around the point of interest in the form of a high-dimensional vector. By establishing the corresponding points in the feature space, F2S3 is also sensitive to the displacement along the surface to produce a complete 3D displacement vector.

The two methods based on point cloud distance and displacement field are the most commonly used methods in surface slope monitoring. The method based on point cloud distance is used more frequently than the method based on displacement field. Except for C2C and M3C2 that directly compare point clouds, other methods all need to establish different types of models. This requires selecting appropriate models for different monitoring scenarios. Simple triangular meshes can be established for those with small fluctuations, while polynomial fitting is often used for those with surface irregularity. At present, there is a lack of unified evaluation of model accuracy, in addition, there is not much research that gives appropriate type correspondence for selecting fitting methods. The displacement field technology does not need to face this problem. The core of the displacement field technology is the matching of point cloud features. Through sliding windows, roughness, and other point cloud features, it is possible to match the differences in features of point clouds at different periods. The key is the method of feature extraction, which is more difficult than point cloud distance measurement. From the above analysis, it can be seen that the average accuracy of TLS in surface deformation monitoring is basically at the centimeter level. This is because the surface deformation area is large, and the point cloud data quality decreases with the increase of distance when TLS scans over long distances.

## 3. Research and Applications in Constructions Deformation Monitoring

### 3.1. Deformation Monitoring of Dam

As an important part of river flood control projects, dams realize the reasonable regulation and optimal allocation of water resources through water storage, which is very important to produce electricity, water supply, and irrigation. They play an important role in the lifeblood of society and economy [40]. However, with the progression of time and the evolution of the natural environment, some dams have serious safety hazards. The collapse of a dam will cause serious threats to the safety of people’s lives and property, and social and ecological damage. Therefore, the safety of dams is crucial for the normal operation of water engineering systems. Continuous monitoring is important in order to prevent any hazardous effects of dams [41].

A reconstruction algorithm based on spherical projection was proposed [42], which used the triangulation algorithm to construct a 3D surface model. The test vector algorithm was used to gradually search for the best local vector along the surface of the object in a similar forward wave manner and propagated down until all points had the correct vector. At the same time, a partial spherical projection unit was generated at each point to express the discrete sampling points and the geometric information of the contained local neighborhood. The overall dam deformation was obtained by comparing the cloud data of different reference points with the reference surface model. NURBS (non-uniform rational B-splines) technology was used to model the point cloud data of the dam and create a high spatial resolution model of the earth rock dam with sufficient accuracy (±2 mm) [43]. As the model was created from a series of continuous monitoring exercises at different times, the deformation characteristics of the earth rock dam are explained more completely. Others used the point cloud registration method to obtain dam displacement. Two-step point cloud registration and contour model comparison method was applied to an arch dam displacement change detection, which can reach millimeter level accuracy [44]. A customized processing procedure for deformation extraction was proposed [45], the core idea of which was to achieve high accuracy registration of the point cloud in an iterative manner using the ICP algorithm for different time periods in the reference region, displaying the features and undeformed geometry at stable locations at each time period. Then, the alignment transformation matrix was applied to the point clouds of each upstream surface in each period, and the multiple adjusted point clouds were compared pairwise for deformation evaluation. The minimum detectable deformation was in the range of below ±10.0 mm if the numerical errors of surface generation were removed. A registration algorithm based on a normal distribution was proposed [46], which improved the original progressive encryption triangulation filter algorithm. Researchers used the C2M method to compare and analyze the patterns of the monitored dam shape variables, and developed a source program to realize the processing, comparative analysis, and modeling of the integrated monitoring point cloud data. The mean square error in dam subsidence displacement was about ±1.98 mm.

In dam deformation monitoring, deformation information should be extracted from the model rather than the point cloud. First, a complete dam model was constructed from the point cloud, including triangulation model, contour model, and function fitting model. Since the point cloud data will inevitably encounter the situation of leakage holes and noise points, it was necessary to interpolate the point cloud data during modeling, which reduced the influence of point cloud roughness. At the same time, the question of whether the point cloud obtained by interpolation was consistent with the actual surface condition deserved consideration. As a means to obtain deformation information, the accuracy of the model affected the accuracy of the final deformation monitoring results directly. The current stage is to explore the improvement of model accuracy, from TIN, the linear contour model, and then to the function fitting model. The high accuracy of the dam surface model is the key goal we pursued. Due to the dam being monitored over a smaller area, the accuracy in deformation monitoring of the dam was at millimeter level.

### 3.2. Deformation Monitoring of Shield Tunnel

With the development and utilization of urban underground space, tunnels are becoming more and more common. Laser scanning technology for deformation monitoring in the tunnel engineering application has caused some concern [47], especially for shield tunnels, which are formed by the assembly of tube pieces, which are regular in shape and facilitate deformation monitoring of their whole using point cloud data. As a key to the construction and maintenance of tunneling projects, traditional monitoring technology only measures a small number of data points, which is not sufficient to understand the deformation of the entire tunnel [48]. With the gradual maturity of TLS technology, when used for tunnel surface survey, the overall deformation distribution of the tunnel can be obtained, overcoming the shortcomings of the traditional survey. The main difficulties include: (1) tunnel central axis extraction, (2) tunnel cross-section extraction, (3) tunnel 3D model construction, (4) water leakage, segment dislocation, and crack identification.

Aiming at the problem of how to effectively obtain 3D models and accurately extract the feature information of large-scale composite structures such as tunnels, a surface-based nondestructive measurement method was proposed and developed, and its accuracy falls within the millimeter range [49]. The method focused on extracting feature sections and central curves for tunnel deformation monitoring. The innovation of this method lies in the projection and iterative filtering of the ring data and rasterization of the point clouds for vertical and horizontal lines. The standard deviation of distances between the central points and central curve was about 3.5 mm. A new method was proposed to extract the tunnel central axis based on the 3D invariant moments and best-fit ellipse, and it can be with an accuracy of 2 mm for cross-section measurement [50]. To enable fast registration of the point cloud, a method of locating the base was proposed, and an improved moving least-square method was proposed to reconstruct the tunnel center line from the unorganized point cloud. After fitting an optimal circle, the cross-sections were estimated by the proposed method. The RMSE of the TLS method was estimated as 4.7 mm. The convergence of the tunnel cross-section was analyzed based on each point cloud slice to determine the safety state of the tunnel [51]. Since the original point cloud collected by TLS cannot show tunnel deformation, a 3D modeling method based on the elliptical fitting algorithm (EFA) was proposed to analyze the settlement deformation, segment dislocation and cross-section convergence of the tunnel. Compared with the results of numerical simulation, the maximum error of the convergence deformation was about 1 mm [52]. To extract the shield tunnel cross-sections from point clouds, a new framework was proposed [53]. It consisted of two steps: tunnel central axis extraction and cross-section determination. A slice-based method was proposed to obtain the initial central axis, which was further divided into linear and nonlinear segments. The circular segment algorithm based on enhanced random sample consensus (RANSAC) tunnel axis segmentation solved the problem of linear and nonlinear hybrid segment extraction. The cross-sectional fitting error was only 1.69 mm. Compared with the designed radius of the metro tunnel, the RMSE of extracted cross-section radius only keeps 1.6 mm. A high-precision interpolation and filtering algorithm was introduced to extract the continuous tunnel profile of the entire tunnel, which was further refined by solving the constrained nonlinear least-squares method. In view of the difficulties in extracting cross-section information and the lack of applicable deformation analysis based on the point cloud, an adaptive cross-section extraction algorithm for deformation analysis was proposed and the deformation analysis accuracy was less than 3 mm [54]. Boundary points were extracted along the point cloud route direction, and the central axis was determined using a bidirectional projection algorithm. After a comprehensive analysis of the curve fitting algorithm, the cubic B-spline curve was selected to fit the cross-section points. Finally, the radial and diametric divergence were used to analyze the local deformation position and overall deformation trend. The technique of acquiring the 3D shape of tunnel surface was proposed for the first time [55], and the 3D deformation reconstruction technology method was based on the ellipse fitting algorithm, coordinate transformation, and the M3C2 algorithm. There are many methods to detect the cracks, water leakage, and other diseases in the shield tunnel. A tunnel surface parameterization algorithm based on a harmonic map was proposed, which reconstructed a triangle mesh model of the tunnel and then generated a harmonic map depth map of the tunnel inner wall on the triangle mesh [56]. It is able to obtain the spatial distribution and location information of the appendages and detect whether there are cracks, water leakage, falling pieces, and other diseases by the depth images. An automatic tunnel monitoring method based on the image data collected by the motion vision measurement unit composed of camera arrays was studied [57], combined with the deep learning algorithm to identify cracks through the Mask R-CNN network. Using the reflection intensity value of the point cloud data, the combination of expansion and the Canny algorithm can realize the automatic intelligent identification and extraction of fractures [58]. A new method for underground tunnel leakage detection was proposed [59]; the reference target was used to correct the influence of distance and incident angle on the intensity based on piecewise linear interpolation. After corrections of distance and incident angle effects, the corrected intensity data were used to detect the water leakage regions in the underground tunnels.

In recent years, the application of TLS technology in engineering surveys has become a technology for large-scale applications in the tunnel engineering environment, due to its advantages of non-contact, fast speed, high precision, and large-scale data acquisition. In order to obtain satisfactory scanning data, this section briefly introduces the application in tunnel monitoring. In tunnel deformation monitoring work, the first thing to do is to extract the central axis, and then slice the tunnel point cloud to fit the cross-sectional analysis. In tunnel deformation analysis, the accurate extraction of the axis is particularly important, and how to fit the central axis accurately and quickly is the key research direction. From the above research, it can be seen that the accuracy of TLS in tunnel axis extraction and section fitting is very high, at the millimeter level. In addition, the geometric deformation of the tunnel is monitored during excavation, and the accuracy and reliability of the 3D reconstruction of the tunnel scene are also to be studied. The real scene in the tunnel is 3D, and the regular monitoring modeling comparison is to determine the deformation trend of the tunnel. Moreover, the recognition of tunnel cracks or leakage areas should also rely on deep learning and image recognition algorithms, after all, image recognition algorithms are more mature, and point cloud intensity information can also be used to monitor water leakage and cracks. TLS has not yet become a common tool for tunnel measurements, but it still has great potential for exploitation.

### 3.3. Deformation Monitoring of Tall Constructions

Some scholars have used TLS to monitor the deformation of some tall constructions. The most common characteristics of these constructions is that they are very tall. Most tall constructions have complex multi-story structures, such as cooling towers, wind turbine towers, and chimneys. The cross-sections of these constructions are usually circular or elliptical. The long-term influence of external loads on these structures causes different degrees of deformation of the structure. This not only affects the stability of the construction itself, but also threatens the security of people’s lives and property. Monitoring the deformation of tall constructions can do a good job of risk transfer and early warning before the risk occurs. Therefore, it is important to monitor the deformation of tall constructions.

Cross-sectional method: surveyors generally use the line scanning mode of observation to fit circles at different heights, and to fit the center coordinates of circles of thin point cloud slices at multiple levels. By comparing the center coordinates of the upper levels with those of the lowest levels, the 2D displacement deviations of the tower moving in the axial and lateral directions can be determined. Some researchers used TLS to conduct line scans of cooling towers, wind turbine towers, and chimneys to analyze axial deformation monitoring [60], and used a least-squares circle fitting method [61,62] to fit a circle from cross-sectional surveying points in the same horizontal plane, the center of the circle and the deviation of the circle were calculated, a comparison of TLS, and classical deformation measurement of two chimneys in a similar way was presented [63]. By continuously measuring the coordinate values on the horizontal plane, the deflection of the object axis can be calculated, and the deviation from the center is the deviation from the vertical line of the measured object. To monitor the verticality of chimneys, two methods were proposed to estimate the center at different levels [64]. The first solution is a manual method using traditional CAD software, where the circular fitting is performed manually by point cloud slicing. The second method is to automatically fit the circle using the least-squares method, which provides not only the central coordinates but also statistics to assess the metric quality and shows a precision better than 2 mm. In monitoring the verticality of a wind power tower and the deformation monitoring of the wind turbine blade, the point cloud data was processed using Bentley MicroStation V8i software for the graphical representation and geometrical measurements [65]. The measurements were used to determine the basic dimensions of the examined facility, deflection of the vertical axis of the stationary support, and blade geometry. The point cloud of the chimney was cut into horizontal rings of 25 mm width by CloudCompare software, vertical cylindrical segments to each ring were fitted by the least-squares method, and then the *x*, *y* coordinates and average adjustment errors of the cylindrical axes could be calculated. By comparing the *x* and *y* coordinates of the higher layer with those of the lowest layer, the deviation of the chimney axis from the vertical direction was calculated [66]. For the assessment of industrial chimney geometry, TLS and structure from motion (SFM) were integrated with an average error of 13 mm, after the registration of point clouds obtained by the two methods, the center of the circle was piecewise fitted to obtain the axis deviation [67].

Surface parametric modeling: Gauss–Helmet nonlinear model was used to estimate the surface parameters of a cylinder for 3D parametric modeling [68]. This method is much more complex than the traditional cross-sectional method in determining the inclination angle of the structure, but it is more advantageous in obtaining the inclination angle of the structural axis directly, avoiding blind spots between the cross-sections, and performing advanced analysis of structural deformation compared with the modeling surface. By establishing a 3D model of the building point cloud data, the inclination of the building can be monitored [69], comparing the angle between the *y* axis of the building’s central axis and the horizontal *x* axis, and determining whether the building is inclined due to ground subsidence by the degree of the angle. An angle of 90° indicates no inclination and less than 90° indicates that the building has been inclined.

In addition, some scholars have used not only TLS, but also combined it with other technologies to monitor the deformation of tall constructions. Based on the TLS technology, the video measurement was also used [70], with the camera placed face up at the bottom of the generator, video observation of the nacelle using point tracking software to measure the movement of the nacelle, and the laser scanner tilted upward to measure the deformation at the top end of the tower. The measurements were compared in time and frequency domains under different operating conditions, such as low/strong winds and turbine start/brake. There was a high correlation between the laser-based signal and the reference measurement signal in the time domain, and the same peak of the main tower oscillation was determined in the frequency domain. The TLS technology was proved to be a capable tool for the structural health monitoring of wind turbine towers. The GB-RAR (ground-based real aperture radar) interferometer can also be used in combination with the TLS for coordinate positioning [71], and the high resolution of the TLS and the advantages of GB-RAR can be used to capture and measure the vibration frequency of the higher oscillation mode to monitor the wind tower. The joint use of co-located TLS and GB-RAR can provide richer information on the dynamical behavior of the structure. Model-based processing is based on the use of interpolation to reduce noise and improve the accuracy of displacement monitoring [72]. After the joint registration of TLS and GB-RAR, the vibration spectra obtained by TLS and GB-RAR can be compared by spectral analysis of distance–time series. This method is especially suitable for linear structures such as wind towers. The oscillation profile can be estimated, the spectral characteristics of the tower can be adequately described, the vibration amplitude of the wind tower can be described, and the structural deformation monitoring of the wind tower can be ensured.

The TLS technology with rapid survey, large number of points, and high recognition accuracy is very effective in surveying the geometry of objects that change rapidly over time. However, its limitation is the repeatability, which requires the identification of new points on the surface each time. The advantage of the cross-section method is the simplicity of the algorithm and its wide application. However, its limitation is that it is not global monitoring, but uses the coordinate difference of the center of the fitting circle of the point cloud cross-section at different heights to determine the axial deviation of the object, and does not use all the point cloud data. The advantage of the surface parametric modeling method is that it directly obtains the inclination angle of the structure axis, avoids the blind spots between the cross sections, and provides a more advanced analysis of the structural deformation compared with the modeling surface. Both methods are more effective when the shape of the monitored object is regular and can be expressed by functions. In addition, other means (such as video measurement, amplitude measurement, etc.) can be combined to achieve more comprehensive monitoring when monitoring the deformation of tall constructions.

## 4. Key Issues in the Application of TLS Technology

### 4.1. Data Acquisition

As numerous studies and applications have been conducted in the field of deformation monitoring, data acquisition is the premise and basis of deformation analysis. It includes inspection of instruments, field survey, control network layout, and scanning plan formulation. It is necessary to check whether the scanner can work normally and whether its accuracy can meet the engineering requirements; then, to go to the site to survey the terrain. For different terrain conditions, the subsequent deployment of the control network and determination of the scanning position of the scanner should be considered in advance. After investigating the terrain, the control network can be established, which is not necessary but recommended. Control points can improve the registration accuracy, which is conducive to transform the local coordinate system of the point cloud into the unified coordinate system. Finally, the scanning plan is formulated, and operators acquire the point cloud data according to the scanning plan.

### 4.2. Data Preprocessing

In the application of TLS technology, data preprocessing is also an extremely important aspect [73]. The first stage of point cloud processing is preprocessing. As part of the processing of the point cloud, the data must be prepared for further use. The two main steps are: registration and filtering. When acquiring point cloud data from TLS, the deformation monitoring range is usually large. It is necessary to scan the monitoring area using multiple stations to obtain the complete point cloud data of the deformation monitoring area. Since the coordinate system of each station is independent, we need to convert the scanning data of different stations to the same coordinate system, and then perform registration operation on the point clouds of different stations. The point clouds of different stations are spliced together to form a complete regional point cloud, which is convenient for later analysis of deformation using point clouds of different periods [74]. In addition, point cloud data may contain some noise points [19,22] due to the occlusion of surrounding objects and the influence of the instrument’s own factors. In addition, the volume of point cloud data is extremely large, and there will be millions or even tens of millions of points in the scanning data of one station. Data redundancy is not conducive to the subsequent deformation analysis. First, the point cloud should be filtered to remove noise points and reduce data redundancy, to meet the data volume of deformation analysis and reduce the consumption of data processing time.

In deformation monitoring projects, multi-station splicing of point clouds in the same period is usually done through white balls or checkerboard markers [15,21], which are placed in the overlapping area scanned twice by the scanner, and then point cloud splicing is carried out by identifying markers through the built-in commercial software. In addition, the algorithm can be used to register point clouds. Registration is divided into two categories: pairwise registration and multi-view registration [75]. Among them, the pairwise registration method is divided into two steps: coarse registration and fine registration. Coarse registration methods include the following categories: hand-selected features, 4PCS (4-points congruent sets), probabilistic registration, and deep learning registration. Fine registration is divided into ICP and NDT (normal distributions transform) and their improvements. Multi-view registration methods are divided into sequential registration-based methods and joint registration-based methods. Point cloud registration is also a hot research topic, and many scholars have proposed different registration algorithms in recent years. For cases where the local registration methods rely on sufficient initialization or converge easily to local minima and the global registration methods rely on the accurate extraction of geometric primitives, a CPD (coherent point drift) algorithm using geometric information and structural constraints for point cloud registration is proposed [76]. The algorithm considers the survey geometry and the intrinsic characteristics of the scene to simplify points and incorporates geometric information as well as structural constraints in the probabilistic model to optimize the so-called matching probability matrix, improving the efficiency and robustness. A novel 3D registration framework was proposed that transformed the point cloud into a mid-level structure space, designed a robust method to find the effective original combination corresponding to the 6D pose of the original point cloud, and then constructed a descriptor based on the hybrid structure [77]. The registration was completed by matching descriptors and calculating rotation and translation parameters. The whole process of this method was performed in the structure space. The advantage of this method is that it can capture geometric structures and semantic features in a larger domain, and it can robustly and effectively reconstruct urban, semi-urban, and indoor scenes. An improved registration method based on a point cloud voxel grid was proposed [78]. The voxel grid structure and index structure of point cloud were established to eliminate the uneven density of the point cloud. Then, based on the distribution of point clouds in the voxel grid, the key points were calculated to represent the whole point cloud. RANSAC was used to find the quaternary basis in the target point cloud according to the overlap rate and the size of the point cloud range. The corresponding quaternion candidate set was determined by matching the affine transformation ratio, and the candidate transformation matrix was calculated by SVD (singular value decomposition) for each candidate quaternion. By choosing a suitable point cloud registration method, the efficiency and accuracy of deformation monitoring can be improved.

In the process of laser scanning, point clouds of unwanted objects such as vegetation and electric wires will inevitably be collected and treated as noise points, which will affect deformation monitoring and analysis [59,79]. Correct filtering of noise points and outliers has also become a prerequisite for the credibility of deformation monitoring results. A new LiDAR point cloud filter [80] was proposed, which was based on mathematical morphological geodesic transformation, and has the advantages of simple concept and easy implementation by morphological filtering method. Compared with the traditional morphological transformation, the geodesic transformation only uses the basic structural elements and converges after a finite number of iterations, which can effectively filter the non-ground points. A multiscale noise removal overall filtering algorithm was proposed [81]. According to the comparison of surface change factors, the point cloud was divided into mutation area and flat area. Finally, the statistical filtering algorithm was used to remove large scale noise in the flat area, and the bilateral filtering algorithm was additionally used to remove small scale noise in the mutation area. In addition to the self-developed algorithm, the point cloud processing software integrated various filtering algorithms to process the data scanned by the corresponding manufacturer’s instruments.

As a common step in all deformation monitoring projects, the registration and filtering of point clouds are sufficient to prove their importance in deformation monitoring studies. The registration speed and accuracy are important indicators for evaluating the registration method, and they are also the directions of future research. Faster point cloud registration can reduce the preprocessing time of point clouds, and higher registration accuracy can build a model that is more consistent with the surface and obtain higher shape variables. Point cloud filtering is required to improve the degree of automation and the accuracy. Nowadays, some projects still use the manual method to remove noise, which undoubtedly increases the time of data processing. How to completely and accurately remove noise points to ensure that the real surface data is not removed is a key issue. Currently, many algorithms can meet the basic requirements, but the presence of noise point outliers is fatal for more precise deformation monitoring projects. It is also a factor restricting the application of TLS in high-precision deformation monitoring. In the future, the filtering algorithm should be developed in a fast and accurate way to promote the application of TLS technology.

### 4.3. Data Quality Assurance

Millimeter accuracy is often required in the deformation monitoring industry. Therefore, data quality is one of the most important factors for the effective use of TLS in the deformation monitoring industry. As with traditional techniques, TLS is subject to different sources of uncertainty during surveying. In this case, it is crucial to identify potential sources of errors which affect the data quality and assess their impact on the results.

When using TLS, the accuracy of point cloud data can be affected by instrument mechanisms, atmospheric conditions, object surface properties, and scan geometry [82,83], and other systematic and random errors can also lead to noise. Many studies have addressed error analysis and performance evaluation of laser scanners, which are essential to ensure adequate data quality and reliability. Several factors affecting accuracy are discussed, including laser incidence angle, object height, surface material, and point cloud density [79]. The larger the scanning distance and incident angle, the greater the influence on the scanning quality. The higher the point cloud density is, the more accurate the deformation results will be. At the same time, the disadvantage is that the data volume becomes larger and the scanning time increases. The impact of the incidence angle on the point cloud data was tested, and it was found that about 80% of the data would be lost when the laser incidence angle was greater than 70 degrees [84]. The presence of high incidence angles in TLS measurements has limited the capability to identify the significant displacement of the targets [85]. Systematic and random errors affect the coordinates of each point in the point cloud, which is directly related to the data quality and subsequent processing of the point cloud. The errors affecting the quality of point cloud data are classified, and the document standard summary is formulated for the performance evaluation of TLS, and the performance methods for evaluating TLS are summarized [86]. A robust target-based TLS self-calibration method was proposed at the algorithm level [87]. The solution was obtained by normalizing the residual vector and calculating the method based on the equivalent covariance matrix, which effectively eliminated the random errors and gross errors related to the observation, and improved the point accuracy from the centimeter to millimeter level. The point cloud registration errors that affect the final deformation estimation were studied [88], appropriate segmentation methods for extracting deformed surface data points were determined, methods for identifying undeformed or reference surfaces for deformation estimation were investigated, and a method for minimizing outliers, noisy data, and/or mixed pixels that affect the deformation estimation was proposed. In addition to the analysis of the influence of the errors, there are also studies to put forward the corresponding error model to offset part of the influence of errors. An error model for TLS measurement was proposed [89], where the error was estimated based on the distance to the object and the incident angle. A stochastic model for TLS observations was proposed in reference [90]. The current error model was extended by classifying atmospheric parameters as randomly correlated basic errors. Others integrated the external models related to atmospheric refraction, beam drift, and incident angle into the seven parameter similarity transformation model to introduce the combined model to detect external errors and record multiple scans [91]. Instrument specification, plane residual accuracy, required point spacing, target color, and target gloss are also factors which users should consider when selecting scanning positions [92]. Moreover, two methods were applied for radiometric correction of laser scanning intensity data [93]. It should be noted that there is still a lack of standardized tests for quantifying the impact of various error sources, as well as uniform accuracy requirements. Therefore, the quality and reliability of acquired point cloud data will still face challenges.

In the future, the impact of various errors on the accuracy of point cloud data should be solved from following two aspects. Firstly, research and development of the TLS equipment should be conducted to improve the scanning distance of the TLS, reduce the scanning limitation caused by the incidence angle, improve the scanning distance and accuracy of the TLS, and select a laser that is less affected by environmental factors as the light source. Secondly, the error correction models should be studied, and the algorithm model should be applied to improve the point cloud data quality. The error models of atmospheric refraction, point cloud reflection, and other influencing factors should be established. The correction model of temperature and humidity should be improved through experiments and analysis.

## 5. Summary and Prospect

With the development of TLS technology, the proportion of its application in deformation monitoring has gradually increased. In this paper, we reviewed the latest development of deformation monitoring based on TLS, especially the deformation monitoring methods used in the fields of surface, tunnel, dam, and tall constructions. We summarized the differences in the methods and the focus of extracting deformation information in different domains. In ground surface and slope monitoring, there are two methods based on point cloud distance measurement and characteristic displacement. Most dam monitoring methods are based on the difference of models. In tunnel monitoring, we should pay attention to the extraction of the central axis, segment dislocation, and leakage cracks. In tall constructions monitoring, the cross-sectional method and surface parametric modeling are used to analyze their tilt and offset. Then, we presented the main steps of point cloud preprocessing, including registration and filtering, and analyzed the factors that affect the quality of point cloud data, as well as the factors limiting the development of TLS were discussed. Finally, we can see from the above research that the accuracy of the TLS is unfixed for different monitoring objects. When monitoring the surface displacement or settlement, the observation range is often large and some places are far away from the scanner, resulting in incomplete point cloud data, which reduces the final overall accuracy. This is the limitation of the application of TLS. However, when monitoring in a small range, objects can be scanned in detail, and the accuracy can mostly be within the millimeter level, so TLS can play an efficient role in short distance and small range observation.

In order to promote the development and application of TLS in deformation monitoring, this paper analyzes and proposes the following future research directions:

(1) Improve the accuracy of the data acquired by the instrument. From the above analysis, we can know that TLS acquires very detailed data in short distance and small ranges, but with the increase in the observation distance, data quality becomes more and more poor, which limits the widespread application. Instrument manufacturers should develop more precise scanners suitable for remote measurement to adapt to deformation monitoring in different scenarios.

(2) Improve the data processing capability. Although a lot of research has been conducted on data processing, there is still a considerable gap between the existing technology and its application requirements. There is great potential in improving the processing efficiency, robustness, and automation of algorithms. In addition, the developed algorithms should be oriented to practical applications and have universal applicability in the field of deformation monitoring. Finally, a standard system should be established to evaluate the performance of the algorithms in order to select appropriate methods for different applications.

(3) Establish a high-precision point cloud model. When TLS is used for deformation monitoring, the scanning point cloud modeling is used to obtain the overall information of the object surface. When the point cloud data is lost [94], the selection of different modeling methods and the precision of the model establishment are particularly important [95]. Five categories are presented to establish a preliminary systematization of these methods: (I) point-based models: single points; (II) point-cloud-based model: point cloud; (III) surface-based model: grid structure; (IV) geometric-based model: continuous surface; (V) parameter-based model: parameters that approximate the surface [96]. An overview of research activities dealing with the modelling of point clouds with regard to the derivation of deformations is provided [97]. It is subdivided according to the measured objects and conveys an application-orientated state-of-the-art areal deformation analysis of these objects. The current application fields are systematically summarized to provide modeling reference for different research fields.

(4) Artificial intelligence (AI) is adopted. At present, AI has been widely used in all walks of life. As a branch of computer science, it has great advantages in processing and analyzing a large amount of data. There is no doubt that AI will become one of the main trends in the field of deformation monitoring in the future. In recent years, various AI techniques, especially deep learning, have been found in previous studies proving its great potential in object detection and quality assessment [98,99,100]. In the future, problems such as project planning, scanning planning, and deformation prediction can be solved with the assistance of AI. In this case, it is urgent to generate point cloud training datasets for deformation monitoring, train network models, and apply them to the deformation monitoring industry.

(5) Use BIM (building information modeling) technology. It is also a trend of point cloud application. Through the construction of building model information, the characteristics of objects and their surrounding environment can be intuitively seen. To achieve an integrated project, i.e., integrate and manage each project phase and management (surveying, architectural design, restoration project, the design of the different installations, the construction project, resource management, management of the result) a BIM approach becomes indispensable [101]. In addition, BIM draws attention from the field of heritage documentation and conservation and has generated a new issue of heritage/historic building information modeling (HBIM) [102]. The HBIM model is no longer just a virtual representation and geometric reconstruction of heritage, but the sub-elements have become advanced objects with rich information, including the quantitative and qualitative description and strict relationship information [103]. The advantages of detailed point cloud data can be combined with its BIM/HBIM to produce a broader application prospect.

(6) Combined with VR/AR (virtual reality/augmented reality) technology. In the current workflow of using point cloud data for architectural applications, the acquisition of point cloud data is usually separated from the visualization and processing of point cloud data. Due to process fragmentation, engineers are unable to obtain meaningful construction activity information from point cloud data in time. Its delay prevents the timely identification and processing of on-site problems, especially those related to geometric quality inspection and construction progress tracking. Due to this limitation, future research needs to develop methods for real-time visualization and processing of point cloud data for practical applications. One possible solution is to combine point cloud acquisition and processing tools with VR/AR technology [104]. Taking the deformation monitoring of building components as an example, once the point cloud data of building components are obtained, the point cloud data can be input into the VR/AR system for visualization, where the point cloud data of different periods can be compared and the deformation differences between them could be viewed. In this way, engineers can visualize and identify geometric quality problems in the field after obtaining point cloud data, and immediately take necessary measures to solve the problem [105]. Future research needs to develop methods for real-time visualization and processing of point cloud data for architectural applications, which can be combined with VR and AR technologies.

## Figures and Tables

**Figure 1 sensors-22-09179-f001:**
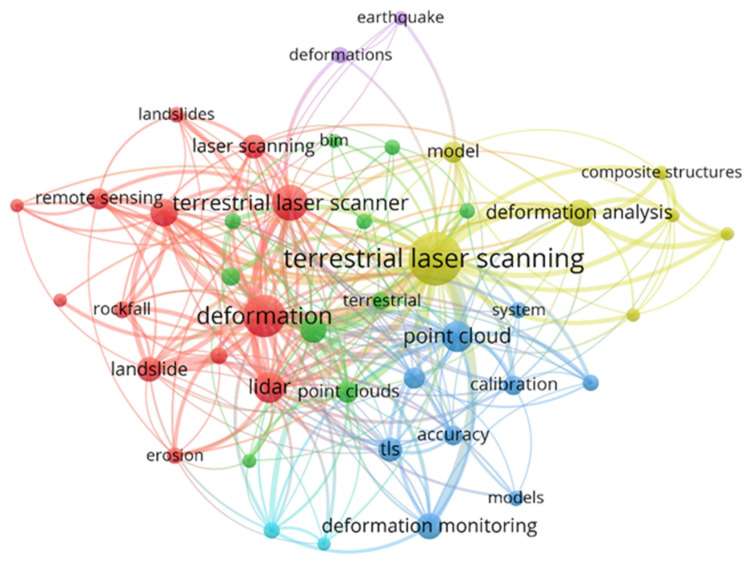
Network of co-occurring keywords.

**Table 1 sensors-22-09179-t001:** Comparison of point cloud ranging methods.

Method	Reference	Advantage	Disadvantage
M2M	[19,20,21]	Direct comparison of terrain models, simple and easy to understand.	Point clouds with rough surfaces cannot be finely meshed, and it is difficult to describe rough and vertical surfaces, which are essentially limited to 2D surfaces.
C2C	[12,16,22,23]	Simple and fast, direct comparison of 3D point clouds, no need to mesh and planarize point clouds, and no need to calculate surface normal.	Sensitivity to point cloud roughness, outliers, and point spacing, and dependence on spatial sampling rate.
C2M	[13,24,25]	Good for flat surfaces.	For rough and occluded point clouds, mesh creation is inconvenient, and interpolation of missing data introduces uncertainties that are difficult to quantify.
M3C2	[14,17,26,28,29]	It is more robust to deal with missing data of point cloud, does not need to mesh the point cloud, and can reduce the influence of noise and outliers in the point cloud.	Long calculation time, relies on the average position in the reference point cloud and data point cloud to reduce noise, insensitive to deformation of flat areas.

## Data Availability

The data presented in this study are available on request from the corresponding author.

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
