# Peer review of "(untitled)"

_sensors, 2022, doi:10.3390/s22239179_

Round 1

Reviewer 1 Report (New Reviewer)

In the manuscript entitled “Review on the Researches and Applications of TLS in Ground Surface and Constructions Deformation Monitoring”, the authors presented a detailed and in-depth review of terrestrial laser scanners technology and its possible applications in ground surface deformation and constructions deformation monitoring. After reading the manuscript, I find the manuscript well written, the discussions about various methods seem reasonable and convincing. The authors did correctly point out some of the key issues in TLS applications and proposals in general to overcome the issues. Overall, as a review paper, the manuscript is good in its present form for acceptance.

An optional suggestion: Table 1 seems to lack a bit qualitative comparison between different methods, e.g., how fast/slow, what is the typical calculation times and so on. Would it be possible to include such details at all?

Author Response

We really appreciate your suggestion. Generally speaking, the calculation time of various methods in Table 1 should be taken as one of the comparison items. However, due to the different complexity of terrain, flat or steep terrain has a great impact on the speed of data processing. The calculation time is also affected by the size of the deformation area, the larger the monitoring area, the larger the amount of point cloud data, and the longer the processing time will be. The purpose of this article is to let readers choose the appropriate method according to the advantages and disadvantages of the method, because our ultimate goal is only to obtain the deformation without too much consideration of time, so we did not add in the end. But in future studies, we will compare the calculation time of various methods in specific monitoring areas based on your suggestion.

Reviewer 2 Report (New Reviewer)

The paper is very interesting and describes many aspects of TLS applications to civil engineering and geotechnical problems.

Just a few corrections of detail:

1) I would not call the SAR more "classical" than TLS perhaps we could call them equally innovative

2) I would specify better that for the study of deformations unfortunately the TLS generally cannot measure the same point twice as opposed to other techniques

3) I would be more cautious to define the TIN difference as a DOD because in general there are no coincident points

4) I would also mention BIM and HBIM applications such as:

Yang, X.; Lu, Y.-C.; Murtiyoso, A.; Koehl, M.; Grussenmeyer, P. HBIM Modeling from the Surface Mesh and Its Extended Capability of Knowledge Representation. ISPRS Int. J. Geo-Inf. 2019, 8, 301.

Angelini, M.G.; Baiocchi, V.; Costantino, D.; Garzia, F. Scan to BIM for 3D reconstruction of the papal basilica of saint Francis in Assisi In Italy. Int. Arch. Photogramm. Remote Sens. Spat. Inf. Sci.—ISPRS Arch. 2017, 42, 47–54.

Fryskowska, A.; Stachelek, J. A no-reference method of geometric content quality analysis of 3D models generated from laser scanning point clouds for hBIM. J. Cult. Herit. 201834, 95–108.

Best regards

Author Response

Thank you so much for your suggestion!

The point-by-point response to the reviewer’s comments  is in the PDF file.

This manuscript is a resubmission of an earlier submission. The following is a list of the peer review reports and author responses from that submission.

Round 1

Reviewer 1 Report

The paper presents a review of applications of terrestrial LiDAR to deformation studies on the monitoring of ground surface and constructions. It is well-written and well-organized. A critical analysis is briefly given inside each section and globally discussed and summarized at the end of the manuscript.

I have just two minor remarks to look at:

1) GB-RAR acronym should be explained at its first use;

2) on page 11, line 494: please verify if "non-ground points" means "non-target points".

Reviewer 2 Report

This paper gives a review about the applications of terrestrial LiDAR in ground surface and constructions deformation monitoring, but lack of adequate support material, such as: examples of representative research results, figure, chart, simulation, experiment, test, etc. Also don’t know the team of authors contribution in this territory.

Reviewer 3 Report

In my opinion, this article is not suitable for publication in its current form.

It has to be substantially rewritten.

It seems that the authors have not considered who and what a review article on an applicative question could be used for. It should provide useful information to those who are involved, by research or practice, with the described technique or proposed applications. A quantitative analysis of the precision that can be achieved cannot be missing. Otherwise, the article simply risks being published but not read by anyone. An article should primarily aim to be read by an adequate audience.

For example, if someone intended to monitor a dam, would they find adequate information here or would it be better to search directly for research articles that deal with the topic, in order to see if the accuracy they intend to achieve is really achievable with the proposed technique?

An interested reader does not have the essential information, only the secondary information. A search with suitable keywords could provide similar results. This is not what a review article should aim for.

Furthermore, it seems that the authors do not take into consideration the developments inherent to other techniques such as SfM, DIC, optical flow. There appears to be nothing other than terrestrial LiDAR (I also note that terrestrial laser scanning is a much more common term than terrestrial LiDAR).

The absence of any kind of scheme, drawing, image, workflow does not help the reader.

In any case, the main deficiency is the total absence of quantitative data.

This article, which in its present form is unnecessarily long, could be used as the basis for a rich and adequate article for publication if the deficiencies described are corrected (I also enclose the annotated manuscript). For the current form I recommend manuscript rejection. 
